# Epidemiological and Genomic Analysis of Asymptomatic SARS-CoV-2 Infections during the Delta and Omicron Epidemic Waves in São Paulo City, Brazil

**DOI:** 10.3390/v15112210

**Published:** 2023-11-03

**Authors:** Svetoslav N. Slavov, Alex R. J. Lima, Gabriela Ribeiro, Loyze P. O. de Lima, Claudia R. dos S. Barros, Elaine C. Marqueze, Antonio J. Martins, Maiara Martininghi, Melissa Palmieri, Luiz A. V. Caldeira, Fabiana E. V. da Silva, Giselle Cacherik, Aline L. Nicolodelli, Simone Kashima, Marta Giovanetti, Luiz Carlos Junior Alcantara, Sandra C. Sampaio, Maria C. Elias

**Affiliations:** 1Center for Viral Surveillance and Serological Assessment (CeVIVAS), Butantan Institute, São Paulo 05507-000, SP, Brazil; svetoslav.slavov@hemocentro.fmrp.usp.br (S.N.S.); alex.lima@butantan.gov.br (A.R.J.L.); gabriela.rribeiro@butantan.gov.br (G.R.); loyze.lima@butantan.gov.br (L.P.O.d.L.); claudia.barros@butantan.gov.br (C.R.d.S.B.); ecmarqueze@gmail.com (E.C.M.); antonio.martins@butantan.gov.br (A.J.M.); 2Ribeirão Preto Medical School, Ribeirão Preto 14051-140, SP, Brazil; skashima@hemocentro.fmrp.usp.br; 3Blood Center of Ribeirão Preto, Ribeirão Preto 14051-140, SP, Brazil; 4Health Surveillance Coordination, Municipal Health Department of São Paulo, Coordenadoria de Vigilância em Saúde (COVISA), Secretaria Municipal de São Paulo (SMS SP), São Paulo 01223-010, SP, Brazil; mmartininghi@prefeitura.sp.gov.br (M.M.); melissapalmieri@prefeitura.sp.gov.br (M.P.); luizcaldeira@prefeitura.sp.gov.br (L.A.V.C.); 5Primary Care Coordination Municipal Health Department of São Paulo, Coordenadoria de Atenção Básica, Secretaria Municipal de São Paulo (SMS SP), São Paulo 01223-010, SP, Brazil; fevilanova@prefeitura.sp.gov.br (F.E.V.d.S.); gcacherik@prefeitura.sp.gov.br (G.C.); anicolodelli@prefeitura.sp.gov.br (A.L.N.); 6University of São Paulo, Ribeirão Preto 14051-140, SP, Brazil; 7Institute of Biological Sciences, Federal University of Minas Gerais, Belo Horizonte 31270-901, MG, Brazil; giovanetti.marta@gmail.com (M.G.); alcantaraluiz42@gmail.com (L.C.J.A.); 8Institute Rene Rachou Foundation Oswaldo Cruz, Belo Horizonte 30190-002, MG, Brazil; 9Sciences and Technologies for Sustainable Development and One Health, University Campus Bio-Medico Rome, 00128 Roma, Italy

**Keywords:** asymptomatic infection, SARS-CoV-2, epidemiology, molecular testing, genomic surveillance, Brazil

## Abstract

We examined the asymptomatic rates of SARS-CoV-2 infection during the Delta and Omicron waves in the city of São Paulo. Nasopharyngeal swabs were collected at strategic points of the city (open-air markets, bus terminals, airports) for SARS-CoV-2 RNA testing. Applying the questionnaire, the symptomatic individuals were excluded, and only asymptomatic cases were analyzed. During the Delta wave, a total of 4315 samples were collected, whereas 2372 samples were collected during the first Omicron wave. The incidence of the asymptomatic SARS-CoV-2 infection was 0.6% during the Delta wave and 0.8% during the Omicron wave. No statistical differences were found in the threshold amplification cycle. However, there was a statistical difference observed in the sublineage distribution between asymptomatic and symptomatic individuals. Our study determined the incidence of asymptomatic infection by monitoring individuals who remained symptom-free, thereby providing a reliable evaluation of asymptomatic SARS-CoV-2 carriage. Our findings reveal a relatively low proportion of asymptomatic cases, which could be attributed to our rigorous monitoring protocol for the presence of clinical symptoms. Investigating asymptomatic infection rates is crucial to develop and implement effective disease control strategies.

## 1. Introduction

SARS-CoV-2 infection might take symptomatic or asymptomatic course. The symptomatic infection is presented as a diversity of symptoms ranging from mild to severe needing mechanical ventilation and supportive oxygen related to high lethality. Both forms of SARS-CoV-2 infection bring important information for the pandemic dynamics. One of the most important consequences of the asymptomatic SARS-CoV-2 carriage is that these individuals might act as an efficient source for viral dissemination. For that reason, the estimation of the asymptomatic rates of SARS-CoV-2 carriage has been crucial for the implementation of non-pharmaceutical measures to control the viral spread and mitigation of the pandemic [1]. However, COVID-19 surveillance has been mainly based on symptomatic infection, and the frequency of asymptomatic infections might fluctuate due to variations in study design and reported percentages [2]. Additionally, the viral transmissibility might be enhanced by the presence of mutations located mainly in the RBD domain [3]. For that reason, studies of the circulating virus variants among asymptomatic individuals are of considerable interest.

The city of São Paulo (SP) is the largest Brazilian and Latin American metropolis. This city was an important entry point for different SARS-CoV-2 variants of concern (VOC) during the pandemic [4]. It was also the epicenter of the Brazilian SARS-CoV-2 outbreak, with the highest number of reported COVID-19 cases [5]. This is due to the strategic position of the city regarding the presence of the most important international airports in the country, the largest bus terminals, and its economic position making it a financial hub not only in Brazil but also in Latin America and worldwide.

Therefore, the objective of this study was to evaluate the asymptomatic SARS-CoV-2 infection rate in the city of SP during two distinct pandemic waves (Delta and Omicron VOCs). The collection of samples was performed by specialized professionals at the most important hot spots of the city distributed across all regions of the city like airports, bus terminals, and open-air markets to evaluate the burden of the asymptomatic SARS-CoV-2 infection. Moreover, the differential of this study was that the positive individuals were followed for symptom evolution by regular phone calls. Thus, it was possible to estimate the likely-real rates of asymptomatic infection in the largest Brazilian city.

## 2. Materials and Methods

### 2.1. Study Design, Sample Collection, and Asymptomatic Tracking

The flowchart of the activities of this study is presented in Figure 1A. The nasopharyngeal samples (NPS) collection was performed in two distinct time periods that coincided with the Delta and Omicron VOCs in the city of SP. The sample collection during the Delta wave was performed between 25 November and 17 December 2021, achieving 4315 collected samples from asymptomatic individuals (confidence level of 99% and power of 99.9%). The sample collection during the Omicron wave was conducted between 28 April and 13 May 2022, obtaining 2372 NPS (confidence level of 99% and power of 99.9%). Both sample collections were taken from important regions of the city with high people circulation (transport terminals and airports) and mass gatherings (open-air public markets) across all regions of the city. As shown in Figure 1B, the city was subdivided into macro-regions: (i) East region, the Municipal open-air market of São Miguel Paulista; (ii) South region: Congonhas national airport and Santo Amaro bus terminal; (iii) Central region: the Municipal Market of São Paulo and Brás (a famous spot for clothing trade); (iv) West region: Municipal Market of Pinheiros; and (v) North region: the Tietê bus terminal (which is the world’s second-largest terminal of its kind). All positive individuals were contacted during their quarantine by health authorities to determine if they exhibited symptoms. The contact was performed by phone call or e-mail. Those individuals who were not accessible both ways were visited by employees of the São Paulo prefecture. The applied questionnaire included the following questions: (1) Were any symptoms observed? (2) If yes, write the date when the symptoms started. (3) What type of symptoms were observed: fever, cough, sore throat, headache, shortness of breath, coryza, smell disturbances, taste disturbances, diarrhea, or other symptoms?. (4) Clinical evolution of the symptomatic cases (in-home treatment, hospital admission, non-emergency unit bed, and emergency unit bed). (5) Any of your contacts showed symptomatic evolution and were positive for SARS-CoV-2? (6) Did any agent of the basic healthcare unit contact you?

Individuals with any clinical symptoms (fever, cough, sore throat, headache, shortness of breath, coryza, smell and taste disturbances, or diarrhea) were excluded from the study. Positive individuals were contacted again after the mandatory quarantine period to confirm their asymptomatic status, with a mean follow-up period of 10 days. For this study, we focused on individuals aged 18 years and above and did not include children. 

### 2.2. SARS-CoV-2 Testing and Variant Assignment

The SARS-CoV-2 RNA testing and whole genome sequencing were performed using previously described protocols [6,7,8]. In brief, viral RNA was automatically extracted from 100 μL of NPS suspension (Extracta kit AN viral, Loccus, Cotia, Brazil) using extractor Extracta 32 (Loccus) following the manufacturer’s guidelines. The SARS-CoV-2 RT-qPCR was performed using the Gene FinderTM COVID-19 Plus RealAmp kit (OSang Healthcare Co. Ltd., Gyeonggi-do, Republic of Korea) following the manufacturer’s instructions. The period between nasopharyngeal sample collection and disclosure of the SARS-CoV-2 result (positive/negative) was 24 h. 

We utilized the COVIDSeq kit (Illumina, San Diego, CA, USA) to sequence the complete genomes of SARS-CoV-2 following the manufacturer’s protocol. The resulting reads were then assembled and assigned to a lineage as previously described [9]. Phylogenetic analysis was performed using the Nextstrain v3.0.3 SARS-CoV-2 workflow [10]. In brief, this workflow aligns the input sequences using nextalign and then reconstructs the phylogenetic trees using IQTree v.2.12 [11]. The phylogenetic analysis utilized a total of 23 Delta VOC and 15 Omicron VOC genomes (Appendix A) that were generated as part of the present study, with at least 85% of genome coverage, and were obtained from asymptomatic individuals. Additionally, a set of 318 Delta VOC and 54 Omicron VOC genomes obtained from the São Paulo State, which were generated by the Butantan Network for Pandemic Alert of SARS-CoV-2 Variants, were included as a symptomatic background for the analysis (Appendix A). The selection of samples was conducted with meticulous care to achieve a comprehensive representation of the genome. Our aim was to cover at least 85% of the entire genome while permitting a maximum of 5 single nucleotide polymorphisms (SNPs) per sample. To ensure accuracy and consistency, the sample collection was exclusively conducted during the period between November and December 2021, with the possibility of including samples from early 2022. The obtained phylogenetic trees were edited using ggtree package in R [12]. 

### 2.3. Statistical Methods

This study involved a statistical descriptive analysis of both quantitative and qualitative variables that characterized the incidence of SARS-CoV-2 positive rates in asymptomatic individuals during the first and second sample collections. To compare the PCR cycle thresholds (Cts) between symptomatic and asymptomatic individuals during the first and second sample collections, we employed the Kruskal–Wallis or Wilcoxon test, depending on the data distribution. In addition, we conducted proportion tests based on the age group of asymptomatic individuals during the first and second sample collections, as well as on the sex of the SARS-CoV-2 positive and negative individuals, the region of sample collection, the use of face masks, and the number of received vaccine doses. In order to compare the distribution of SARS-CoV-2 lineages between asymptomatic and symptomatic individuals during the first and second sample collections, we conducted a Fisher´s exact test.

## 3. Results

### 3.1. Analysis of the Incidence and Epidemiological Data of Asymptomatic SARS-CoV-2 Infections

During the Delta VOC wave sampling, we were able to detect 25 asymptomatic cases out of 4315 collected samples (incidence 0.6%), and during the Omicron VOC wave, we detected 20 asymptomatic infections out of 2372 collected samples (incidence 0.8%). The proportion of symptomatic cases during the Delta wave was 46% (21 cases), and during the Omicron wave, it was 50% (20 cases).There was no statistically significant difference in the incidence of asymptomatic infection between the first and second sample collections. The predominant age range of asymptomatic individuals who tested positive for SARS-CoV-2 was between 21 and 59 years during the first sample collection and between 41 and 59 years during the second sample collection, with no statistically significant difference between both collections (*p* = 0.92). During the first sample collection, none of the asymptomatic individuals reported any comorbidities, but in the second collection, three positive individuals reported comorbidities (diabetes, hypertension, and hepatic disease, respectively) (18.75%). In both sample collections, the majority of the asymptomatically infected individuals were female, but without a statistically significant difference between samplings (56% and 55%, respectively, *p* = 0.95). The majority of the asymptomatic individuals in both sample collections reported complete SARS-CoV-2 vaccination (first and second dose), with no statistically significant difference between collections (88% and 90%, respectively, *p* = 0.39). In the first sample collection, a higher proportion (44%) of the positive participants were from the North region of the city of SP, whereas in the second sample collection, a higher proportion (25%) were from the East and South zones of the city of SP, but without a statistically significant difference between samplings (*p* = 0.22).

### 3.2. Molecular and Genotypic Characteristics of SARS-CoV-2 Infection in Asymptomatic Individuals

We evaluated the PCR Cts of the SARS-CoV-2 infection in asymptomatic individuals and the circulating sublineage. The evaluation of the mean Ct showed that there is no statistical difference between the symptomatic and asymptomatic SARS-CoV-2 individuals (Wilcoxon test *p* = 0.57, first collection = 27.4 + 5.8 and second collection 28.4 + 6.1). The same result was obtained within the groups of the symptomatic (Wilcoxon test *p* = 0.53, first collection = 26.9 + 5.8 and second collection 28.5 + 6.0) and the asymptomatic individuals (Wilcoxon test, *p* = 0.96, first collection = 28.0 = 5.9 and second collection = 28.2 + 6.4%).

In order to gain a more precise understanding of the circulating sublineages among asymptomatic individuals, we conducted sequencing and phylogenetic classification of obtained complete SARS-CoV-2 genomes from the two samplings. However, seven genomes were excluded from the analysis due to low genomic coverage and/or lack of follow-up information on the individuals. Therefore, we analyzed 23 complete genomes obtained during the Delta VOC wave and 15 obtained during the Omicron VOC wave. In general, the average coverage of these samples selected for phylogenetic analysis of the delta variant was 99.42%, whereas the average of the omicron variant was 97.67%. As shown in Figure 2A, during the Delta VOC wave, the most common sublineage infecting asymptomatic individuals was AY.99.2 (56.52%), followed by sublineages AY.43.1, AY.43.2, AY.34.1.1, AY.34, AY.43, and AY.43.7. This sublineage diversity was different compared to that obtained from symptomatic patients (the most represented sublineage was AY.34.1.1, followed by the sublineages AY.99.2 and AY.43), which contributed to shaping the Delta VOC in the SP city (Fisher´s exact test, *p* = 0.002501). During the second sampling, conducted during the Omicron VOC wave, the predominant lineage detected was BA.2 (80.0%), followed by sublineages BA.1.1 and BA.2.9.3, and the recombinant variant XAG (Figure 2A). The XAG identified in this study, when compared to 410 other globally isolated XAG genomes, harbors two unique mutations: A8302G and G25088T. The latter induces a non-synonymous mutation V1176F in the Spike protein, which exhibits a frequency of 0.99% among globally sequenced genomes (including all lineages collected from 1 January 2020 to 7 October 2023), corresponding to a total of 154,900 genomes. Within this dataset, the S:V1176F mutation displays a high frequency in P.1 (Gamma VOC) lineage and its sublineages, and it is detected in 82.75% of the samples (https://cov-spectrum.org/explore/World/AllSamples/AllTimes/variants?aaMutations=S%3AV1176F&, accessed on 10 October 2023). Additionally, between 12 April 2022, and 1 July 2022, 21 sequences with the S:V1176F mutation were recorded in Brazil, with a higher prevalence in the BA.2 (19.05%) and P.2 (14.29%) lineages (https://cov-spectrum.org/explore/Brazil/AllSamples/from%3D2022-04-12%26to%3D2022-07-01/variants?aaMutations=S%3AV1176F&, accessed on 10 October 2023). Similar to the Delta VOC wave, the sublineage diversity was different between symptomatic and asymptomatic individuals (the most prevalent sublineage was also BA.2, followed by the sublineages BA.2.23, BA.2.12.1, and BA.2.56) (Fisher´s exact test, *p* = 0.004622).

We also conducted a phylogenetic analysis of the obtained complete SARS-CoV-2 genomes, including the symptomatic background samples generated by the Butantan Network for Pandemic Alert of SARS-CoV-2 Variants (Figure 2B). Based on our analysis, we observed that the asymptomatic samples were randomly interspersed with genomes obtained from symptomatic individuals. The interactive phylogenetic tree can be obtained by accessing the following link: https://nextstrain.org/fetch/api.onedrive.com/v1.0/shares/u!aHR0cHM6Ly8xZHJ2Lm1zL3UvcyFBczlRcWdFZl9kakF4VFdZc3Q3cUJTOW56cDlpP2U9ejRqMklJ/root/content?d=tree,entropy,frequencies&p=full (accessed on 30 October 2023).

## 4. Discussion

In this study, we investigated the incidence of asymptomatic SARS-CoV-2 infections in strategic areas of the SP city and found a low occurrence of such infections while observing sublineage diversity comparable to that determined during the concurrent pandemic waves. A significant advantage of our study was that the sample collection was conducted in strategic areas of the city, and positive individuals were monitored for the development of clinical symptoms, allowing for the exclusion of presymptomatic positive individuals. The incidence of asymptomatic SARS-CoV-2 infections during the Delta wave was slightly lower (0.6%) than that during the Omicron wave (0.8%), as also observed in other studies evaluating asymptomatic infections during both pandemic waves [13,14]. In our study, we used a rigorous selection protocol for the positive cases, including monitoring of symptom appearance by phone calls, emails, home visits, and application of a questionnaire. For that reason, we believe that the sampling bias was significantly reduced. One of the most significant consequences of asymptomatic SARS-CoV-2 infections is the ability of the virus to rapidly spread due to its increased infectivity. The higher rates of asymptomatic infections observed during the Omicron variant of concern were responsible for the rapid dissemination of this variant globally, in combination with its pronounced vaccine escape [15,16]. The vaccine escape of the Omicron variant of concern is one of its most prominent characteristics. The presence of more than 15 mutations in the RBD domain, in addition to deletions and substitutions in the N-terminal portion of the genome, confers the significant neutralization resistance of this variant to antibodies induced by first-generation vaccines and even monoclonal ones. For that reason, and to confer antibody effectiveness to the Omicron VOC infection, the application of booster vaccine doses is required [17]. These factors contributed to the extensive Omicron variant wave observed in Brazil [18]. 

There is considerable variability in the incidence of asymptomatic SARS-CoV-2 infections reported by different studies and in various geographic locations worldwide. Asymptomatic SARS-CoV-2 infection rates have varied significantly from 0.03% to above 15% [19]. However, in a metanalysis investigation, the percentage of asymptomatic infection was calculated to be 35.1% on the basis of all studies that presented sufficient follow-up of the infected patients [2]. For that reason, the assessment of asymptomatic rates may be affected by different factors, such as the inclusion of presymptomatic individuals and the insufficient or absent follow-ups of positive patients [2]. A notable aspect of our study was that we determined the incidence of asymptomatic infection based on monitored individuals who did not develop clinical symptoms. This allowed for a more robust evaluation of asymptomatic SARS-CoV-2 carriage and depended on the duration of the follow-up, which can also be related to the variation in the incidence of asymptomatic infection [2]. In our study, we followed up for a period of 10 days, which is considered an appropriate duration for monitoring asymptomatic cases [20]. Differences in the sensitivity of diagnostic tests, as well as differences in sample collection and processing, may have contributed to the variation in detection rates of SARS-CoV-2 among asymptomatic patients in different studies. 

Our study investigated the distinction in Ct values, which are directly related to the viral load, between symptomatic and asymptomatic patients with SARS-CoV-2 infection. Our findings revealed no significant difference in Cts between these groups. This result was similar to other studies that examine real-time PCR data and show that there are no differences between the expressions of the most important viral genes like RdRp, E, and N [19,21,22]. This observation highlights the crucial role of asymptomatic individuals as a reservoir of infection that significantly contributes to the maintenance of the pandemic [21], including hosting different sublineages, as our study demonstrated. The identification of the recombinant lineage XAG in an asymptomatic individual sample, but not in the symptomatic samples during the Omicron wave, might be attributed to the scarce circulation of the XAG lineage within the State of São Paulo. Other 11 XAG samples were reported in the State of São Paulo coinciding with our Omicron collection period. Nevertheless, there is no information about the city in which they were sampled. Including the XAG lineage detected in this study (sampled on 11 May 2022), only 43 genomes were reported between 12 April 2022, and 1 July 2022 (based on GISAID data collected on 15 May 2023). Interestingly, the mutation G25088T is linked to a non-synonymous mutation leading to amino acid change V1176F that has been related to hospitalizations and severe outcomes in a study considering a database of 4566 genomes linked to follow-up data [23]. 

Such observations underscore the potential impact of public health interventions aimed at controlling the SARS-CoV-2 spread [24]. The reported incidence of the SARS-CoV-2 asymptomatic infection in our study reinforces the importance of non-pharmaceutical measures, such as wearing face masks, maintaining social distancing, and practicing good hand hygiene, to reduce viral transmission. Additionally, our study identified mild SARS-CoV-2 cases during the follow-up of positive patients, which were excluded from the calculation of asymptomatic infection incidence. However, our results emphasize the benefits of vaccination for the clinical outcomes of SARS-CoV-2 infection in our studied group. Furthermore, the variation in the severity of infection in individuals may be influenced by host factors, such as genetic, environmental, gender, and immunological factors, or a combination thereof [2]. Understanding these factors may help identify individuals at higher risk for severe disease and develop tailored prevention and treatment strategies.

While we observed variations in the sublineage distribution percentages between symptomatic and asymptomatic individuals across multiple samplings, our analysis revealed a random interspersion of the analyzed genomes within the phylogenetic tree. These genomes formed clades, with a maximum of two for the Omicron VOC and three for the Delta VOC. Associating specific lineages and their mutations with clinical outcomes presents a challenging task, as they may be linked to divergent health outcomes. Establishing a reliable association necessitates adjustments for individual risk factors, including the presence of comorbidities such as hypertension, obesity, cardiovascular disease, immunosuppression, smoking, and diabetes mellitus. These comorbidities emerge as more substantial predictors of disease severity, hospitalization, and mortality than the influence of SARS-CoV-2 variants [25]. Therefore, we believe that clinical symptomatology might not be related solely to genetic determinants and may also be associated with the host and environmental factors. In the context of the asymptomatic infection, it was recently proposed that individuals carrying the *HLA-B*15:01* allele (which is estimated to be present in approximately 10% of individuals with European ancestry) exhibit a more than two-fold increased likelihood of maintaining an asymptomatic state following viral exposure compared to those lacking this allele. Notably, homozygous individuals for this allele increase the probability of remaining asymptomatic by over eight-fold [26]. While exploring the genetic underpinnings of the asymptomatic population under investigation bears substantial scientific intrigue, it is important to clarify that this particular avenue of inquiry did not align with the primary objectives of our study. Consequently, pertinent genetic data pertaining to this demographic were not ascertained during our interviews. Nevertheless, it is worth noting that the State of São Paulo exhibits a notably substantial proportion of European genetic ancestry [27], which raises the potential presence of the *HLA-B*15:01* allele within the population of São Paulo city. It is imperative to underscore that establishing a direct link between the genetic background and the progression of SARS-CoV-2 infection in Brazil poses a formidable challenge. This is primarily attributed to the exceptional genetic diversity and extensive admixture characteristic of the Brazilian population, rendering it one of the most heterogeneous constituencies globally.

Similar to investigations into asymptomatic rates of SARS-CoV-2, our study carries certain limitations. We deem the most salient limitations to be the following: (i)The substantial geographical expanse of the city: While our SARS-CoV-2 testing facilities were strategically situated in vital commercial hubs within the city, such as open-air markets, bus terminals, and airport terminals, the city’s vast peripheries, notably including densely populated slum areas, were not encompassed within our study. The omission of these areas may have an impact on the reported rates of asymptomatic SARS-CoV-2 infections.(ii)Voluntary participation in the study: It is imperative to acknowledge that our study relied on voluntary participation. Consequently, not all individuals present within the covered areas availed themselves for testing. This, in turn, may have influenced the estimated rates of asymptomatic SARS-CoV-2 infection.

Notwithstanding these limitations, it is crucial to underscore that they do not compromise the integrity of our findings. Instead, they are germane to any study endeavoring to assess asymptomatic SARS-CoV-2 infection within expansive urban settings.

## 5. Conclusions

In this study, we sought to examine the incidence of asymptomatic SARS-CoV-2 infection during two epidemic waves in the largest Brazilian metropolis. Our findings revealed a relatively low proportion of asymptomatic cases, which may be attributed to our rigorous follow-up protocol that included monitoring for the development of clinical symptoms. Investigating the rates of asymptomatic infections is crucial for effective disease control, particularly in high-risk populations. Moreover, the asymptomatic rates of infection are essential for advancing vaccine development and for anticipating the emergence of new variants in future scenarios.

## Figures and Tables

**Figure 1 viruses-15-02210-f001:**
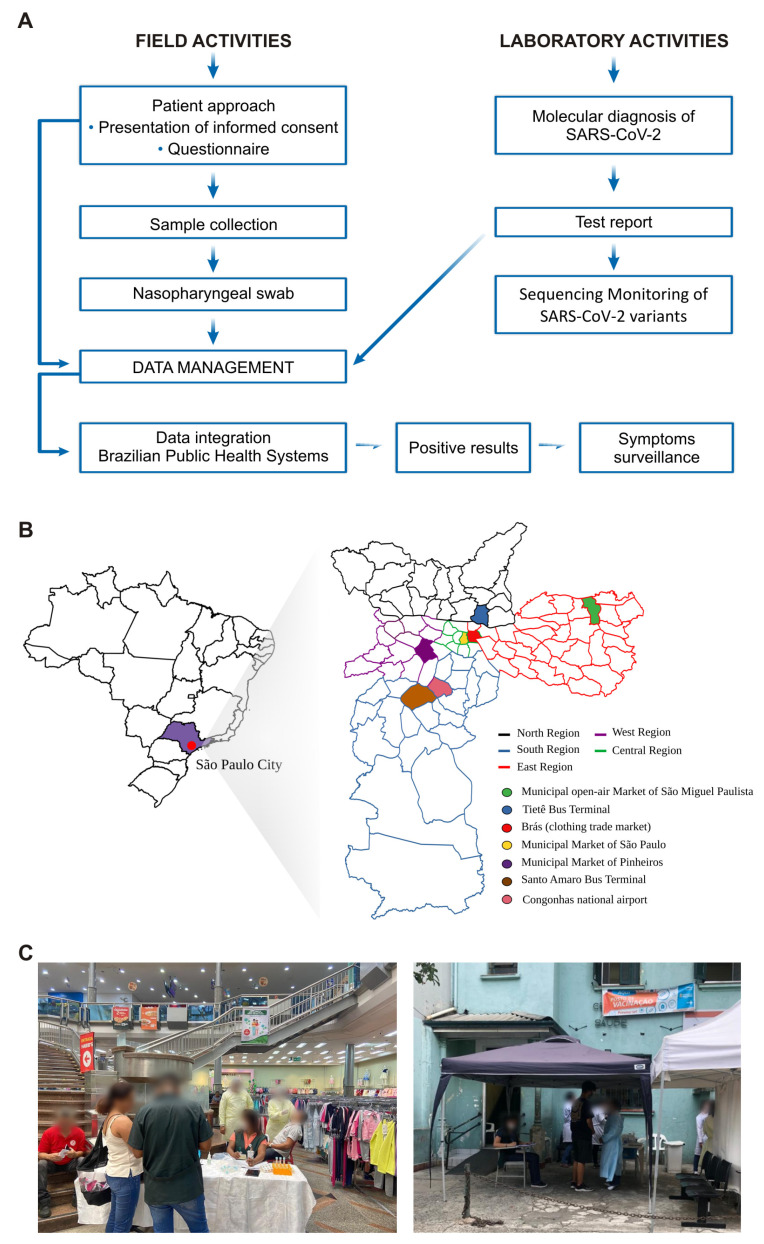
(**A**) Flowchart illustrating the primary field and laboratory procedures involved in assessing the prevalence of asymptomatic SARS-CoV-2 infection in São Paulo city. (**B**) Map of São Paulo, presenting the locations of the sampling points. The map features distinct color-coded lines representing the major city regions, grouped together based on neighborhood proximity. The shaded neighborhoods on the map indicate the precise sampling points chosen for data collection and analysis. (**C**) Image showcasing the collection points for the samples.

**Figure 2 viruses-15-02210-f002:**
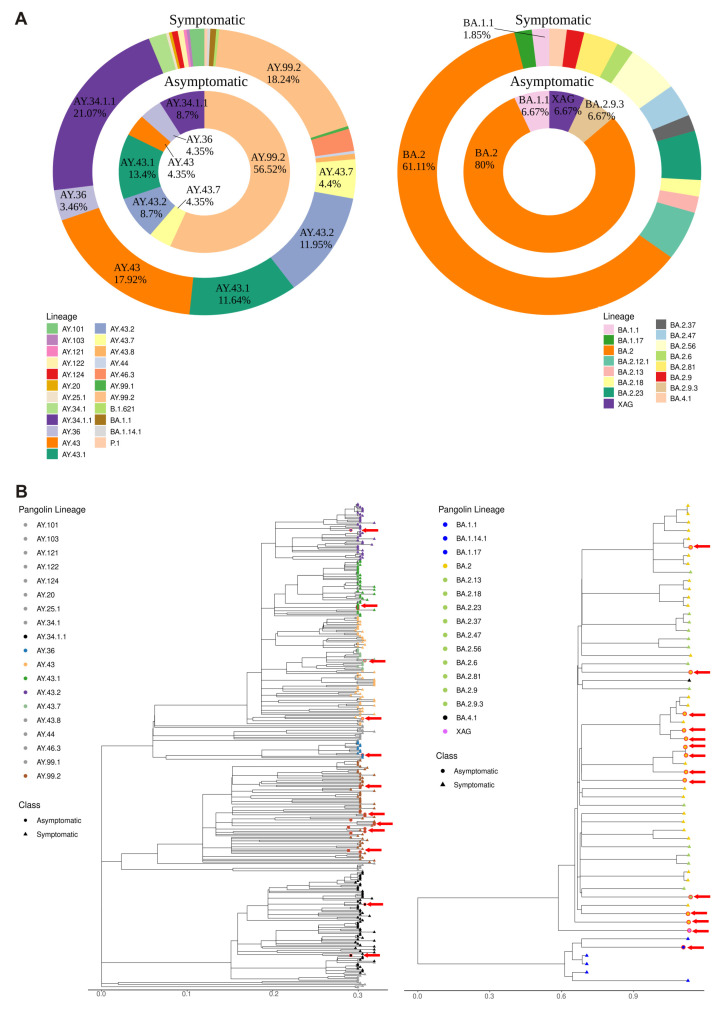
Lineage distribution across samplings. Outer circles: genomes obtained from symptomatic individuals (used as background); inner circles: genomes obtained from asymptomatic individuals (from this study). (**A**—**left**) Lineage distribution from the Delta wave, containing 23 genomes from asymptomatic individuals and 318 genomes from symptomatic individuals. (**A**—**right**) Lineage distribution from the Omicron wave, containing 15 genomes from asymptomatic individuals and 54 genomes from symptomatic individuals. (**B**) Maximum likelihood tree of genomic sequences from São Paulo city. (**B**—**left**) Samples from Delta wave. (**B**—**right**) Samples from Omicron wave. Genomes from symptomatic individuals are represented by triangles in the tip of the trees, while asymptomatic genomes are represented by circles, pointed by red arrows.

## Data Availability

All sequences that were generated and used in the present study are listed in Appendix A, accessible in the GISAID repository, which can be obtained using the respective sequence IDs, sampling dates, the origin and sending laboratories, and the main authors.

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
