# Peer review of "Epidemiological and Genomic Analysis of Asymptomatic SARS-CoV-2 Infections during the Delta and Omicron Epidemic Waves in São Paulo City, Brazil"

_viruses, 2023, doi:10.3390/v15112210_

Round 1

Reviewer 1 Report

Comments and Suggestions for Authors

The paper by Slavov and colleagues is  an epidemiological study of the incidence of asyntomatic SARS-CoV-2 infections in the Sao Paulo area in Brazil.

In general terms, I think the paper is quite well written, and I appreciate its readability and clarity. The genomic analysis is also quite interesting.

I have some perplexities, though, about the study design.

Actually, in the paper there is a frequent confusion, either in the presentation of background knowledge and in the discussion, between two different meanings of 'incidence of asymptomatic infections', one being the proportion of positive cases within the whole asymptomatic population and the other being the proportion of asymptomatic subjects among the total number of positive cases. Studies using one or the other meaning of the term are cited and referred  without distinctions, and this leads the reader into equivocal considerations.

I personally think that establishing the proportion of asymtomatic vs symptomatic cases within the total positive poulation would be of higher interest. Furthermore, in your case, I also have doubts about the possibility of having some biases in the subjects selection, since I'm not so sure that the sampling strategy is adequate for obtaining a population truly representative of the potential target population for SARS-CoV-2 infection. Finally, I can't find, but maybe it's my fault, any account of the symptomatic proportion of the studied cohort, and that would be an interesting information.

So, I would suggest the authors one of the following:

- try to consider a different study design

- optimize the current design by adding relevant information about symptomatic cases, reevaluate the demographics of the cohort to better represent the general population, clearly state the definition of the epidemiological observation they are providing, and rearrange their references paying attention to the noteworthy differences in the studies.

I will propose therefore a major revision

Reviewer 2 Report

Comments and Suggestions for Authors

1.      The study follows a very well-designed methodology. The results may be skewed since patients who showed flu-like symptoms, or any clinical symptoms were excluded. And yet, some of these patients could be "asymptomatic" with respect to the manifestations of SARS-COV-2. This may lead to an excessive elimination of asymptomatic cases with respect to COVID-19, and therefore the ratio of asymptomatic cases may be underestimated.  

2. It would be useful to indicate the content of the questionnaire followed by the health authorities who contacted the positive or suspected cases. Was it a phone call or an email with the questionnaire? Were patients who did not respond at first contact interviewed later or eliminated from follow-up?

3. Could you, please acknowledge potential limitations of the study ?

Comments on the Quality of English Language

Minor editing of English language

Reviewer 3 Report

Comments and Suggestions for Authors

General comments

The authors report about Epidemiological and genomic analysis of asymptomatic SARS- 2 CoV-2 infections during the Delta and Omicron epidemic 3 waves in São Paulo city, Brazil.

While the presented sounds, the interpretation of the collected data may need to be enlarged to include some important literature missed by the authors, that can help readers in the understanding of the presented findings but also authors in considering possible clues that may help in supporting their claims, regarding the possibility to implement effective disease control strategies, as maybe can be argued from the below reported referee report and related concerns.

Introduction and Discussion

In general, the authors report about the proportion of SARS-CoV-2 patients that showed or did not show symptoms. In this concern, it should be appreciated to report since the introduction what the authors mean with “symptomatic” or “asymptomatic”. Which were the biomedical parameters they evaluated for considering an infected individual as symptomatic or asymptomatic.

In addition, they just mentioned the number of people in the two groups affected by the Delta or the Omicron variants. In this regard, it would be useful, if the authors can provide some sentences in the introduction and discussion about relationships between the severity of the symptoms and the varied transmissibility of more recent variants, as well as polymorphisms within the receptors (cited in the below reported papers) considered crucial players in SARS-CoV-2 infections, in the analysed groups also in relation to the different declared ethnicity (considering that they also collected samples from the airport), due to the fact that the ethnicity may reflect a different distribution of variants on ACE2 receptor and other proteins involved in the mechanism of infection.

In this regard the authors may read, cite, and take in consideration for their analysis the below reported papers:

https://pubmed.ncbi.nlm.nih.gov/35013687/

https://pubmed.ncbi.nlm.nih.gov/32946807/

It would also be useful that the authors explain to people not expert in the field how asymptomatic people can favour virus spread. In this regard, the authors may read, cite, and discuss the following paper:

https://pubmed.ncbi.nlm.nih.gov/34287354/

Another concern is related to the employed nomenclature of the investigated variants. In this reviewer opinion the authors should define the variants according to the official names provided by the outbreak database or GISAID database for avoiding confusion. It is important to provide more details about the analysed groups because, as the authors know, there are several sublineages above all in the omicron lineage, which appeared in different periods and/or world regions, along the sample collection performed by the authors. The different sub-lineages should be mentioned in the main text instead of speaking about Omicron and Delta. Notably, variants in the omicron sub-lineages showed mutations that differently impacted on the ability of the virus in spreading through the population, above all compared to the Delta variants. If the authors have data (as mentioned in Figure 1A) about the sequencing of the investigated variants in the two groups, they should provide and briefly discuss those data. I.e., the authors should list preferentially in a dedicated table which were the mutations (i.e., at least those on the RBD?) observed in their biological samples that allowed the authors to identify a specific variant, according to what reported in the below-reported paper to be considered in author discussion and/or in GISAID webpage. It is not sufficient in this reviewer opinion that the authors just mentioned the name of the variant in the reported Table S1. A reference to the different observed virulence and or transmission of the two variants, reflecting the impact of the newly described mutations observed in the two lineages would improve the impact/quality of the manuscript. In this regard, they should read, cite, and discuss the following paper:

https://pubmed.ncbi.nlm.nih.gov/35013687/

METHODS

When the authors report “any clinical symptoms” they should detail which were the clinical symptoms that they considered for excluding an individual from the analysis. They should also explain which were the conditions for considering an individual as asymptomatic after 10 days from the positivity.

It can be useful that the authors provide data about the time employed by asymptomatic.vs. symptomatic individuals in having a PCR result negative to the virus.

The link to the interactive phylogenetic tree does not work.

RESULTS

Letters of Figure 2 panels A-B should be greater (at least those of the panel legends).

DISCUSSION

In the discussion section it is again important to briefly explain what the authors mean with “viral load” estimated through “Cts” (assuming that it is sufficient) of symptomatic versus asymptomatic people (according to the above indicated paper).

In addition, the authors should discuss virulence versus transmissibility already highlighted by other authors in the different investigated variants (according to the above indicated papers).

Concerning vaccine escape the authors just cite references [13, 14]. However, some sentences about the mechanism of infection would help in understanding also the ability of the virus in escaping antibodies. In this regard, the authors may find useful to read, cite, and discuss the following papers

https://pubmed.ncbi.nlm.nih.gov/33127889/

https://pubmed.ncbi.nlm.nih.gov/32817270/

https://pubmed.ncbi.nlm.nih.gov/37338047/

https://pubmed.ncbi.nlm.nih.gov/34149735/

Beyond the statements about errors in the collection of patients samples, the authors should mention possible relationships between the absence of symptoms and polymorphism in human receptors involved in virus infection.

The authors should provide more details about XAG lineage (also through the above-requested Table with sequenced variants).

Comments on the Quality of English Language

English language is ok.

Round 2

Reviewer 1 Report

Comments and Suggestions for Authors

I thank the authors for their explanations.

Nevertheless, I stick on the idea that it would be necessary to also account for the proportion of symptomatic cases among the cohort (I'm sure the authors should have access to this information) along with that of asymptomatics, since the comparison of the two provides a better explanation of the importance of the studies finding, while the mere calculation of the proportion of positive cases among the healthy and socially active  subpopulation (not the general population) of a single urban area is much less informative.

As a second point, please check for accuracy of the data referred on the discussion. In particular, as I had already indicated in the first report, the the proportion of asymptomatic cases in previous studies is provided by considering two completely different meaning of the term, so that the 0,03% is obtained in a study with similarities with the current one, but the 35% absolutely not (in that case, the proportion of asymptomatic among positive subjects, a completely different concept). Please clarify and separate, here and in other possible parts of the text in which a similar misunderstanding could arise.

Reviewer 3 Report

Comments and Suggestions for Authors

The manuscript has been improved according in part to the provided indications. However, this reviewer still believes that the authors should introduce possible relationships with the "genetic background" of the people sample they analysed with reference to the known interactors cited by Singh et al. in Cell Rep 2020.

In addition, some molecular context should be briefly introduced about vaccine escape mechanism, by explaining how and why new variants can escape antibodies related to previous vaccinations and/or infections with previous variants, for making clearer their message to people not expert in the field.

In addition, when this reviewer asked about "the time employed by asymptomatic.vs. symptomatic individuals in having a PCR result negative to the virus", I was asking to the authors to indicate either the " time from the sample collection to the disclosure of the SARS-CoV-2 positive result" or the number of days after which the "patients positive to the virus" had a negative COVID-19 test result". It should be mentioned if the negative result was obtained by nose pharynx swab collection or by PCR.

Comments on the Quality of English Language

Some minor typos need to be corrected
